# Tight Bounds for Volumetric Spanners and Applications

**Aditya Bhaskara**
University of Utah
bhaskaraaditya@gmail.com

**Sepideh Mahabadi**
Microsoft Research–Redmond
smahabadi@microsoft.com

**Ali Vakilian**
Toyota Technological Institute at Chicago (TTIC)
vakilian@ttic.edu

## Abstract

Given a set of points of interest, a volumetric spanner is a subset of the points using which all the points can be expressed using "small" coefficients (measured in an appropriate norm). Formally, given a set of vectors $X = \{v_1, v_2, \ldots, v_n\}$, the goal is to find $T \subseteq [n]$ such that every $v \in X$ can be expressed as $\sum_{i \in T} \alpha_i v_i$, with $\|\alpha\|$ being small. This notion, which has also been referred to as a well-conditioned basis, has found several applications, including bandit linear optimization, determinant maximization, and matrix low rank approximation. In this paper, we give almost optimal bounds on the size of volumetric spanners for all $\ell_p$ norms, and show that they can be constructed using a simple local search procedure. We then show the applications of our result to other tasks and in particular the problem of finding coresets for the Minimum Volume Enclosing Ellipsoid (MVEE) problem.

## 1 Introduction

In many applications in machine learning and signal processing, it is important to find the right "representation" for a collection of data points or signals. As one classic example, in the column subset selection problem (used in applications like feature selection, [Boutsidis et al., 2008]), the goal is to find a small subset of a given set of vectors that can represent all the other vectors via linear combinations. In the *sparse coding* problem, the goal is to find a basis or dictionary under which a collection of vectors admit a sparse representation (see [Olshausen and Field, 1997]).

In this paper, we focus on finding "bases" that allow us to represent a given set of vectors using *small* coefficients. A now-classic example is the notion of an Auerbach basis. Auerbach used an extremal argument to prove that for any compact subset $X$ of $\mathbb{R}^d$, there exists a basis of size $d$ (that is a subset of $X$) such that every $v \in X$ can be expressed as a linear combination of the basis vectors using coefficients of magnitude $\leq 1$ (see, e.g., [Lindenstrauss and Tzafriri, 2013]). This notion was rediscovered in the ML community in the well-known work of Awerbuch and Kleinberg [2008], and subsequently in papers that used such a basis as directions of exploration in bandit algorithms. The term *barycentric spanner* has been used to refer to Auerbach bases. More recently, the paper of Hazan and Karnin [2016] introduced an $\ell_2$ version of barycentric spanners, which they called *volumetric spanners*, and use them to obtain improved bandit algorithms.

The same notion has been used in the literature on matrix sketching and low rank approximation, where it has been referred to as a "well-conditioned basis" (or a *spanning subset*); see Dasgupta et al. [2009]. These works use well-conditioned bases to ensure that every small norm vector (in some normed space) can be expressed as a combination of the vectors in the basis using small coefficients.

37th Conference on Neural Information Processing Systems (NeurIPS 2023).

Recently, Woodruff and Yasuda [2023] used the results of [Todd, 2016] and [Kumar and Yildirim, 2005] on minimum volume enclosing ellipsoids (MVEE) to show the existence of a well-conditioned spanning subset of size $O(d \log \log d)$. (Note that this bound was already superseded by the work of Hazan and Karnin [2016], who used different techniques.) Moreover, [Woodruff and Yasuda, 2023] demonstrated the use of well-conditioned bases for a host of matrix approximation problems (in offline and online regimes) such as low-rank approximation problems, $\ell_p$ subspace embedding, and $\ell_p$ column subset selection.

Our main contribution in this paper is to show that a simple local search algorithm yields volumetric spanners with parameters that improve both lines of prior work by Hazan and Karnin [2016] and Woodruff and Yasuda [2023]. Our arguments also allow us to study the case of having a general $\ell_p$ norm bound on the coefficients. Thus, we obtain a common generalization with the results of Awerbuch and Kleinberg [2008] on barycentric spanners (which correspond to the case $p = \infty$). Further our results can be plugged in, following the same approach of Woodruff and Yasuda [2023], to obtain improvements on a range of matrix approximation problems in these settings.

One application we highlight is the following. Volumetric spanners turn out to be closely related to another well-studied problem, that of finding the minimum volume enclosing ellipsoid (MVEE) for a given set of points, or more generally, for a given convex body $K$. This is a classic problem in geometry [Welzl, 1991, Khachiyan and Todd, 1990]. The celebrated result of Fitz John (e.g., see [Ball, 1992]) characterized the optimal solution for general $K$. Computationally, the MVEE can be computed using a semidefinite programming relaxation [Boyd et al., 2004], and more efficient algorithms have subsequently been developed; see [Cohen et al., 2019]. Coresets for MVEE (defined formally below) were used to construct well-conditioned spanning subsets in the recent work of Woodruff and Yasuda [2023]. We give a result in the opposite direction, and show that the local search algorithm for finding well-conditioned spanning sets can be used to obtain a coreset of size $O(d/\epsilon)$. This quantitatively improves upon prior work, as we now discuss.

## 1.1 Our Results

We start with some notation. Suppose $X = \{v_1, v_2, \ldots, v_n\}$ is a set of vectors in $\mathbb{R}^d$. We say that a *subset $S \subseteq [n]$ is a volumetric spanner* [Hazan and Karnin, 2016] or a *well-conditioned spanning subset* [Woodruff and Yasuda, 2023], if for all $j \in [n]$, we can write $v_j = \sum_{i \in S} \alpha_i v_i$, with $\|\alpha\|_2 \leq 1$. More generally, we will consider the setting in which we are given parameters $c, p$, and we look to satisfy the condition $\|\alpha\|_p \leq c$ (refer to Section 2) for a formal definition. Our main results here are the following.

**Volumetric spanners via local search.** For the $\ell_2$ case, we show that there exists a volumetric spanner as above with $|S| \leq 3d$. Moreover, it can be found in polynomial time via a single-swap local search procedure (akin to ones studied in the context of determinant maximization [Madan et al., 2019]). Furthermore, our approach demonstrates the existence of a volumetric spanner of size $2d - 1$, marking a substantial progress in addressing the question posed by Hazan and Karnin [2016]. This helps to narrow the gap between the obvious $d + 1$ lower bound and their $12d$ upper bound.

In the case of $p = 2$, there are two main prior works. The first is the work of Hazan and Karnin [2016]. They obtain a linear sized basis, similar to our result. However, their result is weaker in terms of running time (by a factor roughly $d^2$), as well as the constants in the size of the basis. Moreover, our algorithm is much simpler, it is simply local search with an appropriate objective, while theirs involves running a spectral sparsification subroutine (that involves keeping track of barrier potentials, etc.) followed by a rounding step.

The second work is the recent result of Woodruff and Yasuda [2023] which utilizes the prior work by Todd [2016]. Their algorithm is simple, essentially employing a greedy approach. However, it incurs an additional $\log \log d$ factor in the size of the spanner due to the analysis of the greedy algorithm. Removing this factor is interesting conceptually, as it shows that local search with an appropriate objective can achieve a stronger guarantee than the best known result for greedy.

**General $p$ norms.** For the case of general $\ell_p$ norms, we show that a local search algorithm can still be used to find the near-optimal sized volumetric spanners. However, the optimal size exhibits three distinct behaviors:

- For $p = 1$, we show that there exist a set $X$ of size $n = \exp(d)$ for which any $\ell_1$ volumetric spanner of size strictly smaller than $n$ can only achieve $\|\alpha\|_1 = \widetilde{\Omega}(\sqrt{n})$. See Theorem 3.8 for the formal statement.
- For $p \in (1, 2)$, we show that $\ell_p$ volumetric spanners that can achieve $\|\alpha\|_p \le 1$ exist, but require $|S| = \Omega\left(\left(\frac{d}{\log n}\right)^{\frac{p}{2p-2}}\right)$. For strictly smaller sized $S$, we show a lower bound akin to the one above for $p = 1$. See Theorem 3.10.
- For $p > 2$, an $\ell_p$ volumetric spanner (achieving $\|\alpha\|_p \le 1$) of size $3d$ exists trivially because of the corresponding result for $p = 2$. See Theorem 3.6 and Corollary 3.12.

Our results show that one-swap local search yields near-optimal sized volumetric spanners for all $\ell_p$ norms. We note that a unified treatment of the $\ell_p$ volumetric spanner (for general $p$), along with matching lower bounds have not been done in any of the prior work. Existing results treated the cases $p = 2$ and $p = \infty$, using different techniques.

**Coresets for MVEE.** While well-conditioned spanning subsets have several applications [Woodruff and Yasuda, 2023], we highlight one in particular as it is a classic problem. Given a symmetric convex body $K \in \mathbb{R}^d$, the minimum volume enclosing ellipsoid (MVEE) of $K$, denoted $\mathsf{MVEE}(K)$, is defined as the ellipsoid $\mathcal{E}$ that satisfies $\mathcal{E} \supset K$, while minimizing $\mathrm{vol}(\mathcal{E})$. We show that for any $K$, there exists a subset $S$ of $O\left(\frac{d}{\epsilon}\right)$ points of $K$, such that

$$\mathrm{vol}(\mathsf{MVEE}(K)) \le (1 + \epsilon)^d \cdot \mathrm{vol}(\mathsf{MVEE}(S)).$$

This result is formally stated in Theorem 4.2. We define such a set $S$ to be a coreset, and while it is weaker than notions of coresets considered for other problems (see the discussion in Section 2.1), it is the one used in the earlier works of Todd [2016], Kumar and Yildirim [2005]. We thus improve the size of the best known coreset constructions for this fundamental problem, indeed, by showing that a simple local search yields the desired coreset. This is conceptually interesting, because it matches the coreset size for a much simpler object, that is the axis-parallel bounding box of a set of points (which always has a coreset of size $\le 2d$, obtained by taking the two extreme points along each axis).

**Other applications.** Our result can be used as a black-box to improve other results in the recent work of Woodruff and Yasuda [2023], such as entrywise Huber low rank approximation, average top $k$ subspace embeddings and cascaded norm subspace embeddings and oblivious $\ell_p$ subsapce embdeddings. In particular, we show that the local search algorithm provides a simple existential proof of oblivious $\ell_p$ subsapce embdeddings for all $p > 1$. In this application, the goal is to find a small size "spanning subset" of a whole subspace of points (i.e., given a matrix $A$, the subspace is $\{x \mid \|Ax\|_p = 1\}$), rather than a finite set. Our results for oblivious $\ell_p$ subsapce embdedding improves the bounds of non-constructive solution of Woodruff and Yasuda [2023] by shaving a factor of $\log \log d$ in size.

## 1.2 Related work

In the context of dealing with large data sets, getting simple algorithms based on greedy or local search strategies has been a prominent research direction. A large number of works have been on focusing to prove theoretical guarantees for these simple algorithms (e.g. [Madan et al., 2019, Altschuler et al., 2016, Mahabadi et al., 2019, Civril and Magdon-Ismail, 2009, Mirzasoleiman et al., 2013, Anari and Vuong, 2022]). Our techniques are inspired by these works, and contribute to this literature.

Well-conditioned bases are closely related to "column subset selection" problem, where the goal is to find a small (in cardinality) subset of columns whose combinations can be used to approximate all the other columns of a matrix. Column subset selection has been used for matrix sketching, "interpretable" low rank approximation, and has applications to streaming algorithms as well. The classic works of [Frieze et al., 2004, Drineas et al., 2006, Boutsidis et al., 2009] all study this question, and indeed, the work of Boutsidis et al. [2009] exploit the connection to well-conditioned bases.

More broadly, with the increasing amounts of available data, there has been a significant amount of work on data summarization, where the goal is to find a small size set of representatives for a data set. Examples include subspace approximation [Achlioptas and McSherry, 2007], projective clustering [Deshpande et al., 2006, Agarwal and Procopiuc, 2003, Agarwal and Mustafa, 2004], determinant

maximization [Civril and Magdon-Ismail, 2009, Gritzmann et al., 1995, Nikolov, 2015], experimental design problems [Pukelsheim, 2006], sparsifiers [Batson et al., 2009], and coresets [Agarwal et al., 2005], which all have been extensively studied in the literature. Our results on coresets for MVEE are closely related to a line of work on *contact points* of the John Ellipsoid (these are the points at which an MVEE for a convex body touches the body). Srivastava [2012], improving upon a work of Rudelson [1997], showed that any convex $K$ in $\mathbb{R}^d$ can be well-approximated by another body $K'$ that has at most $O\left(\frac{d}{\epsilon^2}\right)$ contact points with its corresponding MVEE (and is thus "simpler"). While this result implies a coreset for $K$, it has a worse dependence on $\epsilon$ than our results.

## 2 Preliminaries and Notation

**Definition 2.1** ($\ell_p$-volumetric spanner). *Given a set of $n \geq d$ vectors $\{v_i\}_{i \in [n]} \subset \mathbb{R}^d$ and $p \geq 1$, a subset of vectors indexed by $S \subset [n]$ is an $c$-approximate $\ell_p$-volumetric spanner of size $|S|$ if for every $j \in [n]$, $v_j$ can be written as $v_j = \sum_{i \in S} \alpha_i v_i$ where $\|\alpha\|_p \leq c$.*

*In particular, when $c = 1$ the set is denoted as an $\ell_p$-volumetric spanner of $\{v_1, \cdots, v_n\}$.*

**Determinant and volume.** For a set of vectors $\{v_1, v_2, \ldots, v_d\} \in \mathbb{R}^d$, $\det\left(\sum_{i=1}^d v_i v_i^T\right)$ is equal to the square of the volume of the parallelopiped formed by the vectors $v_1, v_2, \ldots, v_d$ with the origin.

The *determinant maximization problem* is defined as follows. Given $n$ vectors $v_1, v_2, \ldots, v_n \in \mathbb{R}^d$, and a parameter $k$, the goal is to find $S \subseteq [n]$ with $|S| = k$, so as to maximize $\det\left(\sum_{i \in S} v_i v_i^T\right)$. In this paper, we will consider the case when $k \geq d$.

**Fact 2.2** (Cauchy-Binet formula). *Let $v_1, \cdots, v_n \in \mathbb{R}^d$, with $n \geq d$. Then*

$$\det\left(\sum_{i=1}^n v_i v_i^T\right) = \sum_{S \subset [n], |S| = d} \det\left(\sum_{i \in S} v_i v_i^T\right)$$

**Lemma 2.3** (Matrix Determinant Lemma). *Suppose $A$ is an invertible square matrix and $u, v$ are column vectors, then*

$$\det(A + uv^T) = (1 + v^T A^{-1} u) \det(A).$$

**Lemma 2.4** (Sherman-Morrison formula). *Suppose $A$ is an invertible square matrix and $u, v$ are column vectors. Then, $A + uv^\top$ is invertible iff $1 + v^\top A^{-1} u \neq 0$. In this case,*

$$(A + uv^T)^{-1} = A^{-1} - \frac{A^{-1} uv^T A^{-1}}{1 + v^T A^{-1} u}$$

We will also use the following inequality, which follows from the classic Hölder's inequality.

**Lemma 2.5.** *For any $1 \leq p \leq q$ and $x \in \mathbb{R}^n$, $\|x\|_p \leq n^{1/p - 1/q} \|x\|_q$.*

### 2.1 Coresets for MVEE

As discussed earlier, for a set of points $X \subset \mathbb{R}^d$, we denote by $\mathsf{MVEE}(X)$ the minimum volume enclosing ellipsoid (MVEE) of $X$. We say that $S$ is a coreset for MVEE on $X$ if

$$\mathrm{vol}(\mathsf{MVEE}(X)) \leq (1 + \epsilon)^d \cdot \mathrm{vol}(\mathsf{MVEE}(S)).$$

**Strong vs. weak coresets.** The notion above agrees with prior work, but it might be more natural (in the spirit of *strong* coresets considered for problems such as clustering; see, e.g., Cohen-Addad et al. [2021]) to define a coreset as a set $S$ such that for any $\mathcal{E} \supset S$, $(1 + \epsilon)\mathcal{E} \supset X$. Indeed, this guarantee need not hold for the coresets we (and prior work) produce. An example is shown in Figure 1.

## 3 Local Search Algorithm for Volumetric Spanners

We will begin by describing simple local search procedures LocalSearch-R and LocalSearch-NR. The former allows "repeating" vectors (i.e., choosing vectors that are already in the chosen set), while the latter does not.

LocalSearch-NR will be used for constructing well-conditioned bases, and LocalSearch-R will be used to construct coresets for the minimum volume enclosing ellipsoid problem.

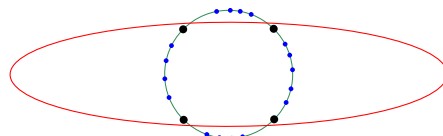

Figure 1: Suppose $X$ is the set of all points (blue and black), and let $S$ be the set of black points. While $\mathsf{MVEE}(X) = \mathsf{MVEE}(S)$, there can be ellipsoids like the one in red, that contain $S$ but not $X$ even after scaling up by a small constant.

---

**Algorithm 1** Procedure LocalSearch-NR

---

1: **Input:** Set of vectors $\{v_1, v_2, \ldots, v_n\} \subseteq \mathbb{R}^d$, parameter $\delta > 0$, integer $r \geq d$
2: **Output:** Set of indices $S$
3: Initialize $S$ of size $r$ using the greedy procedure described in the text
4: Define $M = \sum_{i \in S} v_i v_i^T$
5: **while** $\exists\, i \in S$ and $j \in [n] \setminus S$ such that $\det(M - v_i v_i^T + v_j v_j^T) > (1 + \delta)\det(M)$ **do**
6:     Set $S \leftarrow S \setminus \{i\} \cup \{j\}$
7:     $M \leftarrow M - v_i v_i^T + v_j v_j^T$
8: Return S

---

**Initialization.** The set $S$ is initialized as the output of the standard greedy algorithm for volume maximization [Civril and Magdon-Ismail, 2009] running for $d$ iterations, and then augmented with a set of $(r - d)$ arbitrary vectors from $\{v_1, \ldots, v_n\}$.

**Procedure LocalSearch-R.** The procedure LocalSearch-R (where we allow repetitions) is almost identical to Algorithm 1. It uses the same initialization, however, the set $S$ that is maintained is now a multiset. More importantly, when finding $j$ in the local search step, LocalSearch-R looks over all $j \in [n]$ (including potentially $j \in S$). Also in this case, removing $i$ from $S$ in Line 6 corresponds to removing "one copy" of $i$.

### 3.1 Running time of Local Search

We will assume throughout that the dimension of $\mathrm{span}(\{v_1, v_2, \ldots, v_n\})$ is $d$ (i.e., the given vectors span all of $\mathbb{R}^d$; this is without loss of generality, as we can otherwise restrict to the span).

The following lemma bounds the running time of local search in terms of the parameters $r, \delta$. We note that we only focus on bounding the *number of iterations* of local search. Each iteration involves potentially computing $nr$ determinants, and assuming the updates are done via the matrix determinant lemma and the Sherman-Morrison formula, the total time is roughly $O(nrd^2)$. This can be large for large $n, r$, and it is one of the well-known drawbacks of local search.

**Lemma 3.1.** *The number of iterations of the while loop in the procedures LocalSearch-R and LocalSearch-NR is bounded by*

$$O\left(\frac{d}{\delta} \cdot \log r\right).$$

*Proof.* In every iteration of the while loop, the determinant of the maintained $M$ increases by at least a $(1 + \delta)$ factor. Thus, suppose we define $S^*$ to be the (multi-)set of $[n]$ that maximizes $\det(M^*)$, where $M^* := \sum_{i \in S^*} v_i v_i^T$. We claim that for the $S$ used by the algorithm at initialization (and the corresponding $M$), we have

$$\det(M^*) \leq \binom{r}{d} d! \cdot \det(M). \tag{1}$$

This follows from two observations. First, let $T^*$ be the (multi-)set of $[n]$ that has size exactly $d$, and maximizes $\det(\sum_{i \in T^*} v_i v_i^T)$. Indeed, such a set will not be a multi-set, as a repeated element will reduce the rank. From the bound of Civril and Magdon-Ismail [2009], we have that at initialization, $M$ satisfies

$$\det(\sum_{i \in T^*} v_i v_i^T) \leq d! \cdot \det(M).$$

Next, by the Cauchy-Binet formula, we can decompose $\det(M^*)$ into a sum over sub-determinants of $d$-sized subsets of the columns. Thus there are $\binom{r}{d}$ terms in the summation. Each such sub-determinant is at most $\det(\sum_{i \in T^*} v_i v_i^T)$, as $T^*$ is the maximizer. This proves (1).

Next, since the determinant increases by a factor $(1 + \delta)$ in every iteration, the number of iterations is at most

$$O\left(\frac{1}{\delta}\right) \cdot [d \log d + d \log(er/d)],$$

where we have used the standard bound of $\binom{r}{d} \leq \left(\frac{er}{d}\right)^d$. This completes the proof. $\qquad\square$

## 3.2 Analysis of Local Search

We now prove some simple properties of the Local Search procedures. Following the notation of Madan et al. [2019], we define the following. Given a choice of $S$ in the algorithm (which defines the corresponding matrix $M$), let

$$\tau_i := v_i^T M^{-1} v_i, \quad \tau_{ij} := v_i^T M^{-1} v_j. \tag{2}$$

Note that $M$ is invertible. The determinant of $M$ does not decrease in the local search update (see condition in line 5) and the selected $M$ in the initialization step is invertible. This easily follows from (a) the greedy algorithm is maximizing volume (which is proportional to determinant) and (b) the full set of vectors span a $d$ dimensional space (which is an assumption we can make without loss of generality, as we can always work with the span of the vectors).

We remark that $\tau_i$ is often referred to as the leverage score. We have the following (proof in Section B):

**Lemma 3.2.** *Let $v_1, \ldots, v_n \in \mathbb{R}^d$ and let $S$ be a (multi-)set of indices in $[n]$. Define $M = \sum_{i \in S} v_i v_i^T$, and suppose $M$ has full rank. Then,*

- $\sum_{i \in S} \tau_i = d$,

- *For any $i, j \in [n]$, $\tau_{ij} = \tau_{ji}$.*

The following key lemma lets us analyze how the determinant changes when we perform a swap.

**Lemma 3.3.** *Let $S$ be a (multi-)set of indices and let $M = \sum_{i \in S} v_i v_i^T$ be full-rank. Let $i, j$ be any two indices. We have*

$$\det(M - v_i v_i^T + v_j v_j^T) = \det(M) \left[ (1 - \tau_i)(1 + \tau_j) + \tau_{ij}^2 \right].$$

*Remark.* Note that the proof will not use any additional properties about $i, j$. They could be equal to each other, and $i, j$ may or may not already be in $S$.

*Proof.* By the matrix determinant lemma (Lemma 2.3),

$$\det(M + v_j v_j^T - v_i v_i^T) = \det(M + v_j v_j^T)(1 - v_i^T (M + v_j v_j^T)^{-1} v_i)$$
$$= \det(M)(1 + v_j^T M^{-1} v_j)(1 - v_i^T (M + v_j v_j^T)^{-1} v_i). \tag{3}$$

Next, we apply Sherman-Morrison formula (Lemma 2.4) to get

$$1 - v_i^T (M + v_j v_j^T)^{-1} v_i = 1 - v_i^T \left( M^{-1} - \frac{M^{-1} v_j v_j^T M^{-1}}{1 + v_j^T M^{-1} v_j} \right) v_i = 1 - \tau_i + \frac{\tau_{ij}^2}{1 + \tau_j}. \tag{4}$$

Combining the above two expressions, we get

$$\det(M + v_j v_j^T - v_i v_i^T) = \det(M)(1 + \tau_j) \left[ 1 - \tau_i + \frac{\tau_{ij}^2}{1 + \tau_j} \right].$$

Simplifying this yields the lemma. $\qquad\square$

The following lemma shows the structural property we have when the local search procedure ends.

**Lemma 3.4.** *Let $S$ be a (multi-)set of indices and $M = \sum_{i \in S} v_i v_i^T$ as before. Let $j \in [n]$, and suppose that for all $i \in S$, $\det(M - v_i v_i^T + v_j v_j^T) < (1 + \delta) \det(M)$. Then we have*

$$\tau_j < \frac{d + r\delta}{r - d + 1}.$$

Once again, the lemma does not assume anything about $j$ being in $S$.

*Proof.* First, observe that for any $j \in [n]$, we have

$$\sum_{i \in S} \tau_{ij}^2 = \sum_{i \in S} v_j^T M^{-1} v_i v_i^T M^{-1} v_j = v_j^T M^{-1} M M^{-1} v_j = \tau_j.$$

Combining this observation with Lemma 3.3 and summing over $i \in S$ (with repetitions, if $S$ is a multi-set), we have

$$(1 + \tau_j)(r - \sum_{i \in S} \tau_i) + \tau_j < r(1 + \delta).$$

Now using Lemma 3.2, we get

$$(1 + \tau_j)(r - d) + \tau_j < r + r\delta,$$

and simplifying this completes the proof of the lemma. □

Since Lemma 3.4 does not make any additional assumptions about $j$, we immediately have:

**Corollary 3.5.** *The following properties hold for the output of the Local search procedures.*

1. *For LocalSearch-NR, the output $S$ satisfies: for all $j \in [n] \setminus S$, $\tau_j < \frac{d + r\delta}{r - d + 1}$.*

2. *For LocalSearch-R, the output $S$ satisfies: for all $j \in [n]$, $\tau_j < \frac{d + r\delta}{r - d + 1}$.*

## 3.3 Volumetric Spanners: Spanning Subsets in the $\ell_2$ Norm

We use Lemma 3.1 and Corollary 3.5 to obtain the following.

**Theorem 3.6** ($\ell_2$-volumetric spanner). *For any set $X = \{v_1, v_2, \ldots, v_n\}$ of $n \geq d$ vectors in $\mathbb{R}^d$ and parameter $r \geq d$, LocalSearch-NR outputs a $(\max\{1, \left(\frac{d + r\delta}{r - d + 1}\right)^{1/2}\})$-approximate $\ell_2$-volumetric spanner of $X$ of size $r$ in $O(\frac{d}{\delta} \log r)$ iterations of Local Search.*

*In particular, setting $r = 3d$ and $\delta = 1/3$, LocalSearch-NR returns an $\ell_2$-volumetric spanner of size $3d$ in $O(d \log d)$ iterations of Local Search. Furthermore, setting $r = 2d - 1$ and $\delta = 0$, LocalSearch-NR implies the existence of an $\ell_2$-volumetric spanner of size $2d - 1$.*

*Proof.* Let $S$ be the output of LocalSearch-NR with the parameters $r, \delta$ on $X$. Let $U$ be the matrix whose columns are $\{v_i : i \in S\}$. We show how to express any $v_j \in X$ as $U\alpha$, where $\alpha \in \mathbb{R}^r$ is a coefficient vector with $\|\alpha\|_2$ being small.

For any $j \in S$, $v_j$ can be clearly written with $\alpha$ being a vector that is $1$ in the row corresponding to $v_j$ and $0$ otherwise, thus $\|\alpha\| = 1$. For any $j \notin S$, by definition, the solution to $U\alpha = v_j$ is $\alpha = U^\dagger v_j$, where $U^\dagger$ is the Moore-Penrose pseudoinverse. Thus, we have

$$\|\alpha\|_2^2 = v_j^T (U^\dagger)^T U^\dagger v_j = v_j^T (UU^T)^{-1} v_j = \tau_j.$$

Here we are using standard properties of the pseudoinverse. (These can be proved easily using the SVD). Hence, by Corollary 3.5, we have $\|\alpha\|_2 \leq \left(\frac{d + r\delta}{r - d + 1}\right)^{1/2}$. □

## 3.4 Spanning Subsets in the $\ell_p$ Norm

We now extend our methods above for all $\ell_p$-norms, for $p \in [1, \infty)$. As outlined in Section 1.1, we see three distinct behaviors. We begin now with the lower bound for $p = 1$.

**$\ell_1$-volumetric spanner.** For the case $p = 1$, we show that small sized spanning subsets do not exist for non-trivial approximation factors.

Our construction is based on "almost orthogonal" sets of vectors.

**Lemma 3.7.** *There exists a set of $m = \exp(\Omega(d))$ unit vectors $v_1, \ldots, v_m \in \mathbb{R}^d$ such that for every pair of $i, j \in [m]$, $|\langle v_i, v_j \rangle| \leq c\sqrt{\frac{\log m}{d}}$ for some fixed constant c.*

An example construction of almost orthogonal vectors is a collection of random vectors where each coordinate of each vector is picked uniformly at random from $\{\frac{1}{\sqrt{d}}, \frac{-1}{\sqrt{d}}\}$ (e.g., see [Dasgupta et al., 2009]).

**Theorem 3.8** (Lower bound for $\ell_1$-volumetric spanners). *For any $n \leq \exp(\Omega(d))$, there exists a set of $n$ vectors in $\mathbb{R}^d$ that has no $o(\sqrt{\frac{d}{\log n}})$-approximate $\ell_1$-volumetric spanner of size at most $n - 1$.*

In other words, unless the spanning subset contains all vectors, it is not possible to get an $\ell_1$-volumetric spanner with approximation factor $o(\sqrt{\frac{d}{\log n}})$.

*Proof.* Let $X = \{v_1, \ldots, v_n\}$ be a set of $n$ almost orthonormal vectors as in Lemma 3.7. Suppose for the sake of contradiction, that there exists a spanning subset indexed by $S$ that is a strict subset of $[n]$. Note that for every $i \in [n] \setminus S$ and $j \in S$, $|\langle v_i, v_j \rangle| \leq c\sqrt{\frac{\log n}{d}}$. So, for any representation of $v_i$ in terms of vectors in $S$, i.e., $v_i = \sum_{j \in S} \alpha_j v_j$,

$$1 = \langle v_i, v_i \rangle = \sum_{j \in S} \alpha_j \langle v_i, v_j \rangle \leq \|\alpha\|_1 \cdot c\sqrt{\frac{\log n}{d}}.$$

Hence, $\|\alpha\|_1 \geq \frac{1}{c}\sqrt{\frac{d}{\log n}}$, as long as $|S| < n$. $\qquad\qquad\square$

Note that the lower bound nearly matches the easy upper bound that one obtains from $\ell_2$ volumetric spanners (Theorem 3.6), described below:

**Corollary 3.9.** *For any set of vectors $X = \{v_1, v_2, \ldots, v_n\}$, an $\ell_2$-volumetric spanner is also a $2\sqrt{d}$-approximate $\ell_1$-volumetric spanner. Consequently, such a spanner of size $O(d)$ exists and can be found in $O(d \log d)$ iterations of Local Search.*

The proof follows from the fact that if $\|\alpha\|_2 \leq 1$, $\|\alpha\|_1 \leq \sqrt{3d}$, for $\alpha \in \mathbb{R}^{3d}$ (which is a consequence of the Cauchy-Schwarz inequality). Note that the existence and construction of an $\ell_2$ volumetric spanner of size $3d$ was shown in Theorem 3.6.

**$\ell_p$-volumetric spanner for $p \in (1, 2)$.** Next, we apply the same argument as above for the case $p \in (1, 2)$. Here, we see that the lower bound is not so strong: one can obtain a trade-off between the size of the spanner and the approximation. Once again, the solution returned by LocalSearch-NR is an almost optimal construction for spanning subsets in the $\ell_p$ norm. The proofs are deferred to Section B of the Supplement.

**Theorem 3.10** (Lower bound for $\ell_p$-volumetric spanners for $p \in (1, 2)$). *For any value of $n \leq e^{\Omega(d)}$ and $1 < p < 2$, there exists a set of $n$ vectors in $\mathbb{R}^d$ that has no $o(r^{\frac{1}{p}-1} \cdot (\frac{d}{\log n})^{\frac{1}{2}})$-approximate $\ell_p$-volumetric spanner of size at most $r$.*

*In particular, a (1-approximate) $\ell_p$-volumetric spanner of $V$, has size $\Omega((\frac{d}{\log n})^{\frac{p}{2p-2}})$.*

Next, we show that local search outputs almost optimal $\ell_p$-volumetric spanners.

**Theorem 3.11** (Construction of $\ell_p$-volumetric spanners for $p \in (1, 2)$). *For any set of vectors $X = \{v_1, v_2, \ldots, v_n\} \subset \mathbb{R}^d$ and $p \in (1, 2)$, LocalSearch-NR outputs an $O(r^{\frac{1}{p}-1} \cdot d^{\frac{1}{2}})$-approximate $\ell_p$-volumetric spanner of $X$ of size $r$.*

*In particular, the local search algorithm outputs a 1-approximate $\ell_p$-volumetric spanner when $r = O(d^{\frac{p}{2p-2}})$.*

$\ell_p$**-volumetric spanner for** $p > 2$**.**   The result for $p > 2$ simply follows from the results for $\ell_2$-norm and the fact that $\|x\|_p \le \|x\|_2$ for any $p \ge 2$ when $\|x\|_2 \le 1$.

**Corollary 3.12** ($\ell_p$-volumetric spanner for $p > 2$). *For any set of $n$ vectors $X = \{v_1, v_2, \ldots, v_n\} \subset \mathbb{R}^d$, LocalSearch-NR outputs a $1$-approximate $\ell_p$-volumetric spanner of $X$ of size $r = 3d$ in $O(\frac{d}{\delta} \log d)$ iterations of Local Search.*

## 4   Applications of Local Search and Volumetric Spanners

We now give an application of our Local Search algorithms and volumetric spanners to the problem of finding coresets for the MVEE problem. For other applications, refer to Section A.

**Definition 4.1** (Minimum volume enclosing ellipsoid (MVEE)). *Given a set of points $X = \{v_1, v_2, \ldots, v_n\} \subseteq \mathbb{R}^d$, define $\mathcal{E}(X)$ to be the ellipsoid of the minimum volume containing the points $X \cup (-X)$, where $(-X) := \{-v : v \in X\}$.*

While the MVEE problem is well-defined for general sets of points, we are restricting to sets that are symmetric about the origin. It is well-known (see Todd [2016]) that the general case can be reduced to the symmetric one. Thus for any $X$, $\mathcal{E}(X)$ is centered at the origin. Since $\mathcal{E}$ is convex, one can also define $\mathcal{E}(X)$ to be the ellipsoid of the least volume containing $\text{conv}(\pm v_1, \pm v_2, \ldots, \pm v_n)$, where $\text{conv}(\cdot)$ refers to the convex hull.

As defined in Section 2.1, a coreset is a subset of $X$ that preserves the volume of the MVEE.

**Theorem 4.2.** *Consider a set of vectors $X = \{v_1, \cdots, v_n\} \subset \mathbb{R}^d$. Let $S$ be the output of the algorithm LocalSearch-R on $X$, with*

$$r = \lceil (1 + \frac{4}{\epsilon})d \rceil, \quad \delta = \frac{\epsilon d}{4r}. \tag{5}$$

*Then $S$ is a coreset for the MVEE problem on $X$.*

To formulate the MVEE problem, recall that any ellipsoid $\mathcal{E}$ can be defined using a positive semidefinite (PSD) matrix $H$, as

$$\mathcal{E} = \{x : x^T H x \le d\},$$

and for $\mathcal{E}$ defined as such, we have $\text{vol}(\mathcal{E}) = \det(H^{-1})$, up to a factor that only depends on the dimension $d$ (i.e., is independent of the choice of the ellipsoid). Thus, to find $\mathcal{E}$, we can consider the following optimization problem.

$$(\mathsf{MVEE}) : \min \ - \ln \det(H) \quad \text{subject to}$$
$$v_i^T H v_i \le d \quad \forall i \in [n],$$
$$H \succeq 0.$$

It is well known (e.g., Boyd et al. [2004]) that this is a convex optimization problem. For any $\lambda \in \mathbb{R}^n$ with $\lambda_i \ge 0$ for all $i \in [n]$, the Lagrangian for this problem can be defined as:

$$\mathcal{L}(H; \lambda) = -\ln \det(H) + \sum_{i \in [n]} \lambda_i (v_i^T H v_i - d).$$

Let OPT be the optimal value of the problem MVEE defined above. For any $\lambda$ with non-negative coordinates, we have

$$\text{OPT} \ge \min_H \mathcal{L}(H; \lambda),$$

where the minimization is over the feasible set for MVEE; this is because over the feasible set, the second term of the definition of $\mathcal{L}(H; \lambda)$ is $\le 0$. We can then remove the feasibility constraint, and conclude that

$$\text{OPT} \ge \min_{H \succeq 0} \mathcal{L}(H; \lambda),$$

as this only makes the minimum smaller. For any given $\lambda$ with non-negative coordinates, the minimizing $H$ can now be found by setting the derivative to 0,

$$-H^{-1} + \sum_{i \in [n]} \lambda_i v_i v_i^T = 0 \iff H = \left( \sum_i \lambda_i v_i v_i^T \right)^{-1}.$$

There is a mild technicality here: if $\lambda$ is chosen such that $\sum_i \lambda_i v_i v_i^T$ is not invertible, then $\min_{H \succeq 0}(H; \lambda) = -\infty$. We will only consider $\lambda$ for which this is not the case. Thus, we have that for any $\lambda$ with non-negative coordinates for which $\sum_i \lambda_i v_i v_i^T$ is invertible,

$$\text{OPT} \geq \ln \det \left( \sum_i \lambda_i v_i v_i^T \right) + d - d \sum_i \lambda_i. \tag{6}$$

We are now ready to prove Theorem 4.2 on small-sized coresets.

*Proof of Theorem 4.2.* Let $X = \{v_1, v_2, \ldots, v_n\}$ be the set of given points, and let $S$ be the output of the algorithm LocalSearch-R on $X$, with $r, \delta$ chosen later in (5). By definition, $S$ is a multi-set of size $r$, and we define $T$ to be its support, $\text{supp}(S)$. We prove that $T$ is a coreset for the MVEE problem on $X$.

To do so, define $\text{OPT}_X$ and $\text{OPT}_T$ to be the optimum values of the optimization problem MVEE defined earlier on sets $X$ and $T$ respectively. Since the problem on $T$ has fewer constraints, we have $\text{OPT}_T \leq \text{OPT}_X$, and thus we focus on showing that $\text{OPT}_X \leq d \ln(1 + \epsilon) + \text{OPT}_T$. This will imply the desired bound on the volumes.

Let $S$ be the multi-set returned by the algorithm LocalSearch-R, and let $M := \sum_{i \in S} v_i v_i^T$. Define $\lambda_i = n_i/r$, where $n_i$ is the number of times $i$ appears in $S$. By definition, we have that $\sum_{i \in [n]} \lambda_i = 1$. Further, if we define $H := (\sum_{i \in [n]} \lambda_i v_i v_i^T)^{-1}$, we have $H^{-1} = \frac{1}{r} \cdot M$.

Now, using Corollary 3.5, we have that for all $j \in [n]$,

$$v_j^T M^{-1} v_j < \frac{d + r\delta}{r - d + 1} \implies v_j^T H v_j < \frac{r(d + r\delta)}{r - d + 1} = d \left( 1 + \frac{d - 1}{r - d + 1} \right) \left( 1 + \frac{r\delta}{d} \right).$$

Our choice of parameters will be such that both the terms in the parentheses are $(1 + \epsilon/4)$. For this, we can choose $r, \delta$ as in (5).

Thus, we have that $H' = \frac{H}{(1 + \epsilon)}$ is a feasible solution to the optimization problem MVEE on $X$. This implies that $\text{OPT}_X \leq -\ln \det(H/(1 + \epsilon)) = d \ln(1 + \epsilon) - \ln \det(H)$. The last equality follows because $\det(H/c) = \det(H)/c^d$ for a $d$-dimensional $H$, and any constant $c$.

Next, using the fact that the $\lambda_i$ are supported on $T = \text{supp}(S)$, we can use (6) to conclude that $\text{OPT}_T \geq \ln \det(H^{-1}) = -\ln \det(H)$, where we also used the fact that $\sum_i \lambda_i = 1$.

Together, these imply that $\text{OPT}_X \leq d \ln(1 + \epsilon) + \text{OPT}_T$, as desired. $\qquad \square$

# 5 Conclusion

We show that a one-swap local search procedure can be used to obtain an efficient construction of volumetric spanners, also known as well-conditioned spanning subsets. This improves (and simplifies) two lines of work that have used this notion in applications ranging from bandit algorithms to matrix sketching and low rank approximation. We then show that the local search algorithm also yields nearly tight results for an $\ell_p$ analog of volumetric spanners. Finally, we obtain $O(d/\epsilon)$ sized coresets for the classic problem of minimum volume enclosing ellipsoid, improving previous results by a $d \log \log d$ term.

An interesting open problem arising from our work is that of developing an efficient algorithm to find a $\ell_2$ volumetric spanner of size $2d - 1$. While our local search approach demonstrates the existence of such spanner, it does not find it in polynomial time. Another interesting question is about further improving the gap: is it possible to obtain a volumetric spanner of size smaller than $2d - 1$ for all instances?

## Acknowledgments and Disclosure of Funding

The first author thanks the National Science Foundation for supporting his work via awards CCF-2008688 and CCF-2047288.

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

# A  Other Applications of Volumetric Spanners

We now show some direct applications of our construction of volumetric spanners.

## A.1  Oblivious $\ell_p$ Subspace Embeddings

Oblivious subspace embeddings (OSEs) are a well studied tool in matrix approximation, where the goal is to show that there exist sketching matrices that preserve the norm (say the $\ell_p$ norm) of all vectors in an unknown subspace with high probability. The constructions and analyses of OSEs rely on the existence of a well-conditioned spanning set for the vectors of interest. The following follows directly from Theorem 3.6 (note that we are only using our result for $\ell_2$).

**Theorem A.1** (Improvement of Theorem 1.11 in [Woodruff and Yasuda, 2023])**.** *Let $p \in (1, \infty)$ and let $A \in \mathbb{R}^{n \times d}$. There exists a matrix $R \in \mathbb{R}^{d \times s}$ for $s = 3d$ such that $\|ARe_i\|_p = 1$ for every $i \in [s]$, and for every $x \in \mathbb{R}^d$, $\|Ax\|_p = 1$, there exists a $y \in \mathbb{R}^s$ such that $Ax = ARy$ and $\|y\|_2 \leq 1$.*

The Theorem follows by considering the set

$$X = \{Ax : \|Ax\|_p = 1\},$$

and considering a well conditioned spanning subset in the $\ell_2$ norm. Theorem 3.6 shows the existence of such a subset with $s = 3d$, thus the theorem follows.

However, note that the proof is non-constructive. In order to make it polynomial time, we need to show that the local search procedure can be implemented efficiently. For $p \geq 2$, this may be possible via the classic result of Kindler et al. [2010] on $\ell_p$ variants of the Gröthendieck inequality, but we note that the applications in [Woodruff and Yasuda, 2023] require only the existential statement.

## A.2  Entrywise Huber Low Rank Approximation

The Huber loss is a classic method introduced as a robust analog to the least squares error. A significant amount of work has been dedicated to finding low-rank approximations of a matrix with the objective of minimizing the entry-wise Huber loss. The following offers a slight improvement over [Woodruff and Yasuda, 2023].

**Theorem A.2.** *Let $A \in \mathbb{R}^{n \times d}$ and let $k \geq 1$. There exists a polynomial time algorithm that outputs a subset $S \subset [d]$ of columns in $A$ of size $O(k \log d)$ and $X \in \mathbb{R}^{S \times d}$ such that*

$$\|A - A|^S X\|_H \leq O(k) \min_{rank(A_k) \leq k} \|A - A_k\|_H,$$

*where $A|^S$ denotes the matrix whose columns are the columns of $A$ indexed by $S$ and $\|\cdot\|_H$ denotes the entrywise Huber loss.*

Note that the size of $S$ is reduced from $O(k \log \log k \log d)$ to $O(k \log d)$. The proof of Theorem A.2 follows from Theorem 1.6 in [Woodruff and Yasuda, 2023] and our improved construction for $\ell_2$-volumetric spanner, i.e., $O(1)$-approximate spanning subset of size $O(d)$ (see Theorem 3.6).

## A.3  Average Top $k$ Subspace Embedding

For a given vector $v \in \mathbb{R}^n$, the average top $k$ loss is defined as

$$\|v\|_{\mathrm{AT}_k} := \frac{1}{k} \sum_{i \in [k]} |v_{[i]}|,$$

where $v_i$ denotes the $i$th largest coordinate in $v$.

Using the result of [Woodruff and Yasuda, 2023] relating the problem of average top $k$ subspace embedding to $\ell_2$-volumetric spanners as a black-box, we have the following theorems for small $k$ (i.e., $k \leq 3d$) and large $k$ (i.e., $k > 3d$) respectively.

**Theorem A.3** (small $k$)**.** *Let $A \in \mathbb{R}^{n \times d}$ and let $k \leq 3d$. There exists a set $S \subset [n]$ of size $O(d)$ such that for all $x \in \mathbb{R}^d$,*

$$\|A|_S x\|_{\mathrm{AT}_k} \leq \|Ax\|_{\mathrm{AT}_k} \leq O(\sqrt{kd}) \cdot \|A|_S x\|_{\mathrm{AT}_k},$$

*where $A|_S$ denotes the set of rows in $A$ indexed by $S$.*

**Theorem A.4** (large $k$). *Let $A \in \mathbb{R}^{n \times d}$ and let $k \geq k_0$ where $k_0 = O(d + \frac{1}{\delta})$. Let $P_1, \ldots, P_{\frac{k}{t}}$ be a random partition of $[n]$ into $\frac{k}{t}$ groups where $t = O(d + \log \frac{1}{\delta})$. For every $i \in [\frac{k}{t}]$, there exists a set $S_i \subset N_i$ of size $O(d)$ such that with probability at least $1 - \delta$, for all $x \in \mathbb{R}^d$,*

$$\|A|_S x\|_{\mathrm{AT}_k} \leq \|Ax\|_{\mathrm{AT}_k} \leq O(\sqrt{td}) \cdot \|A|_S x\|_{\mathrm{AT}_k},$$

*where $S := \bigcup_{i \in [\frac{k}{t}]} S_i$ and $A|_S$ denotes the set of rows in $A$ indexed by $S$.*

In both regimes, compared to [Woodruff and Yasuda, 2023], our improved bounds for $\ell_2$-volumetric spanner saves a factor of $\log \log d$ in the number of rows and a factor of $\sqrt{\log \log d}$ in the distortion. The proofs of above theorems respectively follows from Theorem 3.11 and 3.12 of [Woodruff and Yasuda, 2023] together with our Theorem 3.6.

### A.4  Cascaded Norm Subspace Embedding

Next, we explore the implications of the improved bound of $\ell_2$-volumetric spanner (i.e., Theorem 3.6) for embedding a subspace of matrices under $(\|\|\|_\infty, \|\||)$-cascaded norm, which first evaluates an arbitrary norm of the rows and then return the maximum value over the $n$ rows.

The following is a consequence of Theorem 3.13 in [Woodruff and Yasuda, 2023] and our Theorem 3.6. We describe our result for the $(\|\|\|_\infty, \|\||)$-cascaded norm, which first evaluates an arbitrary norm of rows and then return the maximum value over the $n$ rows.

**Theorem A.5** $((\|\|\|_\infty, \|\||)$-subspace embedding). *Let $A \in \mathbb{R}^{n \times d}$ and let $\|\||$ be any norm on $\mathbb{R}^m$. There exists a set $S \subset [n]$ of size at most $3d$ such that for every $X \in \mathbb{R}^{d \times m}$,*

$$\|A|_S X\|_{(\|\|\|_\infty, \|\||)} \leq \|AX\|_{(\|\|\|_\infty, \|\||)} \leq O(\sqrt{d}) \|A|_S X\|_{(\|\|\|_\infty, \|\||)}$$

## B  Missing Proofs

### B.1  Proof of Lemma 3.2

*Proof.* Note that the second part follows from the symmetry of $M$ (and thus also $M^{-1}$). To see the first part, note that we can write $v_i^T M^{-1} v_i = \langle M^{-1}, v_i v_i^T \rangle$, where $\langle U, V \rangle$ refers to the entry-wise inner product between matrices $U, V$, which also equals $\mathrm{Tr}(U^T V)$. Using this,

$$\sum_{i \in S} \tau_i = \sum_{i \in S} \langle M^{-1}, v_i v_i^T \rangle = \langle M^{-1}, M \rangle = \mathrm{Tr}(I) = d.$$

In the last equality, we used the symmetry of $M$. $\qquad\square$

### B.2  Proof of Theorem 3.10

*Proof.* The proof follows from the same argument as before. Consider a set of $n > r$ almost orthonormal vectors $X = \{v_1, \cdots, v_n\} \subset \mathbb{R}^d$ from Lemma 3.7.

Consider an index $i \in [n] \setminus S$ and let $v_i = \sum_{j \in S} \alpha_j v_j$. By Lemma 2.5, for any $p > 1$,

$$\|\alpha\|_p \geq r^{\frac{1}{p}-1} \cdot \|\alpha\|_1 = r^{\frac{1}{p}-1} \cdot \frac{1}{c} \left( \frac{d}{\log n} \right)^{\frac{1}{2}}.$$

In particular, to get a 1-approximate $\ell_p$-volumetric spanner, i.e., $\|\alpha\|_p = 1$, the spanning subset must have size $r = \Omega((\frac{d}{\log n})^{\frac{p}{2p-2}})$. $\qquad\square$

### B.3  Proof of Theorem 3.11

*Proof.* By Corollary 3.5, the local search outputs a set of vectors in $X$ indexed by the set $S \subset [n]$ of size $r > d$ such that for every $i \in [n] \setminus S$, $v_i$ can be written as a linear combination of the vectors in

the spanner, $v_i = \sum_{j \in S} \alpha_j v_j$, such that $\|\alpha\|_2 \le \left( \frac{d+r\delta}{r-d+1} \right)^{\frac{1}{2}}$. By Lemma 2.5 and setting $\delta = d/r$, for any $1 < p < 2$,

$$\|\alpha\|_p \le r^{\frac{1}{p}-\frac{1}{2}} \cdot \left( \frac{d+r\delta}{r-d+1} \right)^{\frac{1}{2}} = O(r^{\frac{1}{p}-1} \cdot (d+r\delta)^{\frac{1}{2}}) = O(r^{\frac{1}{p}-1} \cdot d^{\frac{1}{2}}).$$

In particular, if we set $r = O(d^{\frac{p}{2-2p}})$, the subset of vectors $S$ returned by LocalSearch-NR is an (exact) $\ell_p$-volumetric spanner; i.e., for every $i \in [n] \setminus S$, $\|\alpha\|_p \le 1$.

Finally, the runtime analysis follows immediately from Lemma 3.1. $\qquad\square$

