# Tight Bounds for Volumetric Spanners and Applications

## Abstract

Given a set of points of interest, a volumetric spanner is a subset of the points using which all the points can be expressed using "small" coefficients (measured in an appropriate norm). Formally, given a set of vectors $X = \{v_1, v_2, \ldots, v_n\}$, the goal is to find $T \subseteq [n]$ such that every $v \in X$ can be expressed as $\sum_{i \in T} \alpha_i v_i$, with $\|\alpha\|$ being small. This notion, which has also been referred to as a well-conditioned basis, has found several applications, including bandit linear optimization, determinant maximization, and matrix low rank approximation. In this paper, we give almost optimal bounds on the size of volumetric spanners for all $\ell_p$ norms, and show that they can be constructed using a simple local search procedure. We then show the applications of our result to other tasks and in particular the problem of finding coresets for the Minimum Volume Enclosing Ellipsoid (MVEE) problem.

## 1 Introduction

In many applications in machine learning and signal processing, it is important to find the right "representation" for a collection of data points or signals. As one classic example, in the column subset selection problem (used in applications like feature selection, [Boutsidis et al., 2008]), the goal is to find a small subset of a given set of vectors that can represent all the other vectors via linear combinations. In the *sparse coding* or problem, the goal is to find a basis or dictionary under which a collection of vectors admit a sparse representation (see [Olshausen and Field, 1997]).

In this paper, we focus on finding "bases" that allow us to represent a given set of vectors using *small* coefficients. A now-classic example is the notion of an Auerbach basis. Auerbach used an extremal argument to prove that for any compact subset $X$ of $\mathbb{R}^d$, there exists a basis of size $d$ (that is a subset of $X$) such that every $v \in X$ can be expressed as a linear combination of the basis vectors using coefficients of magnitude $\leq 1$ (see, e.g., [Lindenstrauss and Tzafriri, 2013]). This notion was rediscovered in the ML community in the well-known work of Awerbuch and Kleinberg [2008], and subsequently in papers that used such a basis as directions of exploration in bandit algorithms. The term *barycentric spanner* has been used to refer to Auerbach bases. More recently, the paper of Hazan et al. [2013] introduced an $\ell_2$ version of barycentric spanners, which they called *volumetric spanners*, and use them to obtain improved bandit algorithms.

The same notion has been used in the literature on matrix sketching and low rank approximation, where it has been referred to as a "well-conditioned basis" (or a *spanning subset*); see Dasgupta et al. [2009]. These works use well conditioned bases to ensure that every small norm vector (in some normed space) can be expressed as a combination of the vectors in the basis using small coefficients. Woodruff and Yasuda [2023] used the results of [Todd, 2016] and [Kumar and Yildirim, 2005] on minimum volume enclosing ellipsoids (MVEE) to show the existence of well conditioned spanning subset of size $O(d \log \log d)$. (Note that this bound was already superseded by the work of Hazan et al. [2013], who used different techniques.)

Our main contribution in this paper is showing that a simple local search algorithm yields volumetric spanners with parameters that improve both lines of prior work Hazan et al. [2013] and Woodruff and Yasuda [2023]. Our arguments also allow us to study the case of having a general $\ell_p$ norm bound on the coefficients. Thus, we obtain a common generalization with the results of Awerbuch and Kleinberg [2008] on barycentric spanners (which correspond to the case $p = \infty$). Woodruff and Yasuda [2023] also showed a range of low-rank approximation problems (in offline and online regimes) for which well-conditioned spanning subsets are useful, and our result can be plugged in to obtain improvements in these settings.

One application we highlight is the following. Volumetric spanners turn out to be closely related to another well-studied problem, that of finding the minimum volume enclosing ellipsoid (MVEE) for a given set of points, or more generally, for a given convex body $K$. This is a classic problem in geometry [Welzl, 1991, Khachiyan and Todd, 1990]. The celebrated result of Fitz John (e.g., see [Ball, 1992]) characterized the optimal solution for general $K$. Computationally, the MVEE can be computed using a semidefinite programming relaxation [Boyd et al., 2004], and more efficient algorithms have subsequently been developed; see [Cohen et al., 2019]. Coresets for MVEE (defined formally below) were used to construct well-conditioned spanning subsets in the recent work of Woodruff and Yasuda [2023]. We give a result in the opposite direction, and show that the local search algorithm for finding well-conditioned spanning sets can be used to obtain a coreset of size $O(d/\epsilon)$. This quantitatively improves upon prior work, as we now discuss.

We now present our results in detail.

## 1.1 Our Results

We start with some notation. Suppose $X = \{v_1, v_2, \ldots, v_n\}$ is a set of vectors in $\mathbb{R}^d$. We say that a *subset $S \subseteq [n]$ is a *volumetric spanner* [Hazan et al., 2013] or a *well-conditioned spanning subset* [Woodruff and Yasuda, 2023], if for all $j \in [n]$, we can write $v_j = \sum_{i \in S} \alpha_i v_i$, with $\|\alpha\|_2 \leq 1$. More generally, we will consider the setting in which we are given parameters $c, p$, and we look to satisfy the condition $\|\alpha\|_p \leq c$ (refer to Section 2) for a formal definition.

Our main results here are the following.

**Volumetric spanners via local search.** For the $\ell_2$ case, we show that there exists a volumetric spanner as above with $|S| \leq 3d$. Moreover, it can be found via a single-swap local search procedure (akin to ones studied in the context of determinant maximization Madan et al. [2019]). This improves on the constructions of Hazan et al. [2013], Woodruff and Yasuda [2023] in terms of the size of $S$ obtained. Our result is also simpler, without relying on spectral sparsification or coresets for minimum volume ellipsoids.

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

Next, using the fact that the $u_i$ are supported on $T = \text{supp}(S)$, we can use (5) to conclude that $\text{OPT}_T \geq \ln \det(H^{-1}) = -\ln \det(H)$, where we also used the fact that $\sum_i \lambda_i = 1$.

Together, these imply that $\text{OPT}_X \leq (1 + \epsilon)d + \text{OPT}_T$, as desired. $\qquad \square$

## 5 Conclusion

We show that a one-swap local search procedure can be used to obtain an efficient construction of volumetric spanners, also known as well-conditioned spanning subsets. This improves (and simplifies) two lines of work that have used this notion in applications ranging from bandit algorithms to matrix sketching and low rank approximation. We then show that the local search algorithm also yields nearly tight results for an $\ell_p$ analog of volumetric spanners. Finally, we obtain $O(d/\epsilon)$ sized coresets for the classic problem of minimum volume enclosing ellipsoid, improving previous results by a $d \log \log d$ term.

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

## A  Other Applications of Volumetric Spanners

We now show some direct applications of our construction of volumetric spanners.

### A.1  Oblivious $\ell_p$ Subspace Embeddings

Oblivious subspace embeddings (OSEs) are a well studied tool in matrix approximation, where the goal is to show that there exist sketching matrices that preserve the norm (say the $\ell_p$ norm) of all vectors in an unknown subspace with high probability. The constructions and analyses of OSEs rely on the existence of a well-conditioned spanning set for the vectors of interest. The following follows directly from Theorem 3.6 (note that we are only using our result for $\ell_2$).

**Theorem A.1** (Improvement of Theorem 1.11 in [Woodruff and Yasuda, 2023]). *Let $p \in (1, \infty)$ and let $A \in \mathbb{R}^{n \times d}$. There exists a matrix $R \in \mathbb{R}^{d \times s}$ for $s = 3d$ such that $\|ARe_i\|_p = 1$ for every $i \in [s]$, and for every $x \in \mathbb{R}^d$, $\|Ax\|_p = 1$, there exists a $y \in \mathbb{R}^s$ such that $Ax = ARy$ and $\|y\|_2 \leq 1$.*

The Theorem follows by considering the set

$$X = \{Ax : \|Ax\|_p = 1\},$$

and considering a well conditioned spanning subset in the $\ell_2$ norm. Theorem 3.6 shows the existence of such a subset with $s = 3d$, thus the theorem follows.

However, note that the proof is non-constructive. In order to make it efficient, we need to show that the local search procedure can be implemented efficiently. For $p \geq 2$, this may be possible via the classic result of Kindler et al. [2010] on $\ell_p$ variants of the Gröthendieck inequality, but we note that the applications in [Woodruff and Yasuda, 2023] require only the existential statement.

### A.2  Entrywise Huber Low Rank Approximation

The Huber loss is a classic method introduced as a robust analog to least squares error. There has been a lot of work on finding low rank approximations to a matrix where the goal is to minimize the entry-wise Huber loss. The following slightly improves upon the work of