# OpenReview forum: "Tight Bounds for Volumetric Spanners and Applications"
_NeurIPS.cc/2023/Conference — NeurIPS 2023 poster_

### Official Review · Reviewer_4Fat · 2023-06-22

**Soundness:** 3 good
**Presentation:** 3 good
**Contribution:** 2 fair
**Rating:** 4
**Confidence:** 4

**Summary:**

This paper researches the $\ell_2$-volumetric spanner (or $\ell_2$-well-conditioned spanning subset) for a given dataset $X$ based on the local search algorithm, in which each iteration implements a single swap to update the volumetric spanner $S$. More generally, the results are extended to $\ell_p$-norm for any $p \ge 1$. Furthermore, the proposed algorithm can be applied to build coresets of size $O(d/\epsilon)$, which is independent of the number of data points $n$, for the minimum volume enclosing ellipsoid (MVEE) problem.

**Strengths:**

(1) The proposed algorithm is very easy to understand and the paper is well-written.
(2) This paper presents strict proof for the proposed algorithm.
(3) The coresets for the minimum volume enclosing ellipsoid problem are of size $O(d / \epsilon)$ which is independent of the number of points $n$.


**Weaknesses:**

1. For the $\ell_2$-volumetric spanner case
(1) In the paper [Hazan, Karnin, and Meka'13] (see Theorem 1.1), the  $1$-approximate spanner $S$ has size $|S| = 12 d$ and the running time is $O(n^{3.5} + n^3 d + d^5)$.
(2) In the paper [Woodruff and Yasuda'23] (see Theorem 3.4 and Corollary 3.5), the $(1+\epsilon)$-approximate spanner $S$ has size $|S| = O(d \log\log{d} + d / \epsilon)$ and the running time is $\widetilde{O}((\text{nnz}(A) + d^2) d / \epsilon)$, where $\text{nnz}(A) = O(nd)$.
(3) In this paper, setting $\delta = O(\epsilon)$ gives the $(1+\epsilon)$-approximate spanner $S$ of size $|S| = O(d)$ and the running time $O(n d^4 \log{d} / \epsilon)$.
From the above comparison, we can see that the running time is much worse than that of [Woodruff and Yasuda'23] taking $\epsilon \in (0, 1)$ as a constant.
2. This paper gives the coresets for MVEE with size $|S| = O(d / \epsilon)$ and running time $O(n d^4 \log (d/\epsilon) / \epsilon^2)$. The size of coreset is impressive, but the running time and approximation ratio $(1+\epsilon)^d$ are not satisfactory. In the paper [Cohen, Cousins, Lee, and Yang'19] (see Theorem 1.1), the $\sqrt{d}$-approximate ellipsoid can be obtained in time $O(n d^2 \log (n/d) / \epsilon)$. The coreset is a proxy of dataset $X$, but computing the coreset of $X$ in this paper takes much more time than computing the approximate ellipsoid of $X$ directly in the paper of Cohen et al. For me, this is a critical drawback.

**Questions:**

In Line 37-39, this paper claims that it improves both lines of prior work [Hazan, Karnin, and Meka'13] and [Woodruff and Yasuda'23]. What are the aspects of improvements?

**Limitations:**

See Weaknesses.

---

> ### Author Rebuttal · Authors · 2023-08-09
>
> > For the $\ell_2$-volumetric spanner case
> > (1) In the paper [Hazan, Karnin, and Meka'13] (see Theorem 1.1), the 1-approximate spanner $S$  has size $|S| = 12d$ and the running time is $O(n^{3.5} + n^3d +d^5)$.
> >
> > (2) In the paper [Woodruff and Yasuda'23] (see Theorem 3.4 and Corollary 3.5), the ($1+\epsilon$)-approximate spanner $S$  has size $|S| = O(d \log \log d + d/\epsilon)$ and the running time is $\tilde{O}( (\mathrm{nnz}(A) + d^2) d/\epsilon)$, where $\mathrm{nnz}(A) = O(nd)$.
> >
> > (3) In this paper, setting $\delta = O(\epsilon)$ gives the ($1+\epsilon$)-approximate spanner $S$ of size $|S| = O(d)$ and the running time $O(nd^4 \log d / \epsilon)$. From the above comparison, we can see that the running time is much worse than that of [Woodruff and Yasuda'23] taking $\epsilon \in (0,1)$ as a constant.
>
> For point (1), note that our algorithm (see Theorem 3.6) returns a 1-approximate spanner $S$ with size $3d$ in $O(d \log d)$ iterations (which means $O(nd^4 \log d)$ time even if we do a naive implementation – more on this below). So, both spanner size and runtime in our algorithm are better. Moreover, we don't need to appeal to the machinery on spectral sparsification, barrier potentials, etc.  to obtain our result.
>
> For points (2) and (3), we first note that the greedy algorithm of [Woodruff and Yasuda 23] (henceforth [WY23]) needs to do essentially the same computation as us: they have to compute which vector to add in order to maximize the determinant (i.e., to compute a leverage score). Indeed, [WY23] refers to an earlier work of [Todd] for this step. The $\mathrm{nnz}(A)$ bound is obtained by using sketching methods to speed up the computation of leverage scores. We can use exactly the same methods for quickly computing leverage scores in our local search step. [Todd] needs to perform $d$ steps of greedy addition (each step involving computing appropriate leverage scores, which takes $\mathrm{nnz}(A)$ time), while we need to perform $d \log d$ rounds of local search. Since each round involves (a) removing each of the elements in $S$ one at a time, (b) computing which new vector to add to $S$, similar to greedy addition. So the running time of each local search step is $|S| \times \mathrm{nnz}(A) = O(d \times \mathrm{nnz}(A))$. The number of rounds is larger by a factor $\log d$ as we noted above, so the overall runtime is worse than [WY23] by a factor $d\log d$.
>
> This may be too high a price to pay for improving a $\log \log d$ factor in some applications, but we have the interesting conceptual message that local search gives a strict improvement over the best known guarantee for greedy. Also, our analysis is simple and direct, it does not require appealing to the result of Todd, and the connection to MVEE. (Indeed, in Section 4, we use the connection to MVEE in the opposite direction, obtaining a better result for MVEE.)
>
>
> > This paper gives the coresets for MVEE with size $|S| = O(d/\epsilon)$ and running time $O(nd^4 \log(d/\epsilon) / \epsilon^2)$. The size of coreset is impressive, but the running time and approximation ratio $(1+\epsilon)^d$ are not satisfactory. In the paper [Cohen, Cousins, Lee, and Yang'19] (see Theorem 1.1), the $\sqrt{d}$-approximate ellipsoid can be obtained in time O(nd^2 log(n/d) / \epsilon). The coreset is a proxy of dataset $X$, but computing the coreset of $X$ in this paper takes much more time than computing the approximate ellipsoid of $X$ directly in the paper of Cohen et al. For me, this is a critical drawback.
>
> The work on fast algorithms for John’s ellipsoids *measures approximation differently*. A $\sqrt{d}$-approximation in that context translates to a $(\sqrt{d})^d$-approximation to the volume (because if we scale a body by $c$, the volume in $d$ dimensional space grows by a factor of $c^d$). So in this sense, our construction is much stronger in terms of approximation ratio. [This is a natural question that readers may have, so we will add the above to the final version.]
>
> Second, as we mentioned above, using sketching methods for computing leverage scores, our running time can be brought down to $O(d^2 + \mathrm{nnz}(A))(d^2 \log d) / \epsilon$, which is about $d \log d$ worse.
>
> > Question. In Line 37-39, this paper claims that it improves both lines of prior work [Hazan, Karnin, and Meka'13] and [Woodruff and Yasuda'23]. What are the aspects of improvements?
>
> We described the improvements over prior works in our responses above. We also listed them more clearly in the common rebuttal above (see “Message to all Reviewers”). We will add this discussion to the final version.

---

> > ### Comment · Reviewer_4Fat · 2023-08-14
> >
> > Thank you for your clear responses!
> > For the comparisons with some related work, for example, [Hazan, Karnin, and Meka'13] and [Woodruff and Yasuda'23], it would be more clear if you could use a table to list the size of volumetric spanner, the number of iterations, and running time, etc.
> > In addition, I have the following concerns.
> > (1) For $\ell_2$-volumetric spanner case, comparing with the running time of [WY'23], the running time in your paper is worse by a factor of $d \log{d}$ even though you use sketching techniques to compute leverage scores quickly. Although the proposed method is simple and does not appeal to Todd's result, these cannot convince me.
> > (2) The core part of the proposed method is the Local Search, where the set $S$ is initialized by [Civril and Magdon-Ismail'09] and the subsequent operations are just swaps between $S$ and $\overline{S}$. What's the running time of obtaining the initial $S$? Overall, I think the contribution of this paper is kind of limited.

---

> > > ### Author Response · Authors · 2023-08-16
> > >
> > > >For the comparisons with some related work, for example, [Hazan, Karnin, and Meka'13] and [Woodruff and Yasuda'23], it would be more clear if you could use a table to list the size of volumetric spanner, the number of iterations, and running time, etc.
> > >
> > > Sure, thanks for your suggestion. We will add a more detailed description of our results and how it improves the previous work and will add a table of results too.
> > >
> > > >(1) For $\ell_2$-volumetric spanner case, comparing with the running time of [WY'23], the running time in your paper is worse by a factor of $d \log{d}$ even though you use sketching techniques to compute leverage scores quickly. Although the proposed method is simple and does not appeal to Todd's result, these cannot convince me.
> > >
> > > Note that besides being simple, our local search algorithm improves the size of $\ell_2$- volumetric spanner (aka $\ell_2$-well-conditioned subset) and shaves a factor of $\log \log d$ from the bound of [WY’23] getting an optimal bound upto a constant. We emphasize that the same local search algorithm obtains optimal size for $\ell_p$-volumetric spanners for any $p\ge 1$.
> > >
> > > Moreover, we would like to once again emphasize that our local search approach also provides “optimal” size coreset for MVEE. So, it has applications beyond the work of [WY’23].
> > >
> > > >(2) The core part of the proposed method is the Local Search, where the set $S$ is initialized by [Civril and Magdon-Ismail'09] and the subsequent operations are just swaps between $S$ and $\overline{S}$. What's the running time of obtaining the initial $S$?
> > >
> > > Regarding the running time of the greedy initialization, it is basically the same as [WY’23]. Note that since in local search we need to remove one vector and then greedily add the best one from the remaining vectors, the runtime of local search is a factor of d worse than the greedy initialization per each round and as the local search terminates in $O(d\log d)$ and the greedy algorithm terminates in $O(d)$ rounds, overall the greedy initialization part is faster than the local update part by a factor of $O(d \log d)$ and does not affect the total runtime asymptotically.

---

> > > > ### Comment · Reviewer_4Fat · 2023-08-16
> > > >
> > > > Thank you for your responses!
> > > > 1. Yes, your work improves the $\ell_2$-volumetric spanner size by a factor of $\log\log{d}$ at the cost of a factor of $d \log{d}$ in the running time.
> > > > 2. As you claim that you obtain the optimal $\ell_p$-volumetric spanner size for any $p \ge 1$, I think the main contribution is the lower bounds for $p = 1$ and $p \in (1, 2)$.
> > > > 3. Overall, this paper's contributions lie in giving the (almost) optimal size of $\ell_p$-volumetric spanner for $p \ge 1$ and applying the Local Search method to construct coresets for MVEE problem.
> > > > (1) In terms of contributions, it is acceptable.
> > > > (2) In terms of technique, the proposed method is straightforward and kind of insufficient, that is, given the set $S$ by [Civril and Magdon-Ismail'09], then implementing swap operations between $S$ and $\overline{S}$. Simultaneously, the running time is worse by a factor of $d \log{d}$ compared with the existing work [WY'23].
> > > > Therefore, I retain my score.

---

> > > > > ### Author Response · Authors · 2023-08-19
> > > > >
> > > > > We thank the reviewer for reading our rebuttal and engaging positively. One final comment on algorithmic novelty: we view the fact that local search works as a positive, not as a weakness. The choice of objective and the analysis showing that it works are the main contributions here.
> > > > >
> > > > > In summary, we believe that obtaining _general, tight bounds_ using conceptually simple algorithms _is_ more important than purely improved running times with suboptimal bounds. Of course, we agree that this is subjective.

---

### Official Review · Reviewer_QeGT · 2023-07-06

**Soundness:** 3 good
**Presentation:** 3 good
**Contribution:** 3 good
**Rating:** 6
**Confidence:** 4

**Summary:**

This paper studies the problem of constructing small volumetric spanners. Given a set S of points in R^d, (the l2 version of) the problem is to find a subset of S so that every point in S can be written as a linear combination of the subset points, with the l2 norm of the coefficients bounded above by 1. Analogous problems can be defined for other lp norms.

There is a simple and efficient greedy algorithm that by folklore results gets a spanner of size O(d log d); there is an improvement due to Todd that gets O(d log log d), and Hazan, Karnin, and Meka get 12d via methods involving the John ellipsoid and spectral sparsification. The main result of this paper is that a slight modification of the greedy algorithm actually computes a spanner of size 3d.

**Strengths:**

This paper gives a simpler and (slightly) better algorithm for computing an l2 volumetric spanner: specifically, a natural greedy approach works. It also gives simple extensions to other lp norms, and matching lower bounds.

**Weaknesses:**

The improvement in the spanner size is modest compared to prior work, and the algorithm is quite similar to known approaches (the only difference is that instead of iteratively adding elements to the spanner, each time we add a new element we also remove one), as is the analysis. I would not be entirely sure that this algorithm/analysis is not already present in the literature. However, since I could not find a reference I would err on the side of acceptance.

It would have been good to explicitly mention the bound obtained by the prior work (Hazan, Karnin, and Meka 2013).

**Questions:**

N/A

---

> ### Author Rebuttal · Authors · 2023-08-09
>
> >The improvement in the spanner size is modest compared to prior work, and the algorithm is quite similar to known approaches (the only difference is that instead of iteratively adding elements to the spanner, each time we add a new element we also remove one), as is the analysis. I would not be entirely sure that this algorithm/analysis is not already present in the literature. However, since I could not find a reference I would err on the side of acceptance.
>
> >It would have been good to explicitly mention the bound obtained by the prior work (Hazan, Karnin, and Meka 2013).
>
> Please see the “message to all reviewers” above (common rebuttal). As for the algorithm being known, it indeed is, as it is simply local search! But the key aspect of local search is the choice of the objective function. In this case, we use an appropriate determinant, and we have to argue that the output of local search with this objective is a well conditioned basis.
>
> We also note that the general $\ell_p$ case has interesting behavior which was not known before, and moreover, the exact same algorithm works for all $p$! (Earlier, the cases $p=2$ and $p=\infty$ were handled by different methods).

---

> > ### Comment · Reviewer_QeGT · 2023-08-13
> > **Thanks**
> >
> > Thanks for the response. I concur with the other reviewers about including a comparison table with the prior work, and I retain my score.

---

### Official Review · Reviewer_TY5V · 2023-07-18

**Soundness:** 3 good
**Presentation:** 3 good
**Contribution:** 2 fair
**Rating:** 6
**Confidence:** 1

**Summary:**

This paper looks at the problem of identifying volumetric spanners from a set of points  \in \RR^d. The paper shows a method of local search which can be extended to find near optimal bounds of volumetric spanners, improving on previous algorithms. The authors also apply it to an application called MVEE, where they provide better rates for finding coresets of MVEE than previously in the literature.

The authors start with the generalization of volumetric spanners to the c-approximate use-case and analyze their local search method to show that for an L2 volumetric spanner of size 3d, the local search would only take O(d log d) time. They also extend their results for general l_p cases for p = 1, p \in (1,2) and p>2, the result holding trivially for the latter.

They analyze the coresets of MVEE problem to show that they can obtain coresets of size O(d/\epsilon) using local search method, which is an improvement over previous methods which have O(d/\epsilon^2) bounds.

**Strengths:**

I have not completely checked the proofs, but the results are strong if the proofs hold.
Particularly important is their usage in the spanning subset setting/usecase, which is a frequent need in today's age of very large sized datasets and having a small sized subset spanning the majority of the data points  can open doors to a lot of efficient analysis with higher dimensional data, as well as help us perform matrix vector operations on such datasets more efficiently.
The problem is very easy to describe and motivate and the algorithm is easy to follow.

**Weaknesses:**

The comparison of results is quite difficult to follow from the paper. In particular, it could be pretty easily explained by having a table or having a clear description of the best possible previous result and how much it was improved by through the local search method. I am also not clear about the degree of optimality, which can be more clearly stated in the description itself. It would also be useful to have a description of the degree of optimality of the previous attempts to solve this problem.

**Questions:**

Please look at the weakness section.

---

> ### Author Rebuttal · Authors · 2023-08-09
>
> >The comparison of results is quite difficult to follow from the paper. In particular, it could be pretty easily explained by having a table or having a clear description of the best possible previous result and how much it was improved by through the local search method. I am also not clear about the degree of optimality, which can be more clearly stated in the description itself. It would also be useful to have a description of the degree of optimality of the previous attempts to solve this problem.
>
> We thank the reviewer for this comment. We addressed this in the “message to all reviewers” above (common rebuttal), and we will add the discussion to the final version of the paper.

---

> > ### Comment · Reviewer_TY5V · 2023-08-18
> >
> > I have gone through reply by the authors and the description of the improvements over previous work and I would like to retain my score.

---

### Official Review · Reviewer_JkQ2 · 2023-07-24

**Soundness:** 2 fair
**Presentation:** 2 fair
**Contribution:** 3 good
**Rating:** 6
**Confidence:** 4

**Summary:**

This paper develops and analyzes a local-search algorithm for finding volumetric spanners under norms in the regime where the number of given vectors $n$ is at least as high as the dimension $d$ of these vectors. Moreover, a runtime of the algorithm is given and the size of the algorithm's output is compared to existing lower bounds. Finally, this paper presents applications of the algorithm to the minimum volume enclosing ellipsoid (MVEE) problem (in the paper's body) and other lower rank problems (in the appendices).

**Strengths:**

The main novelty of the paper is the development of a (relatively) simple algorithm that obtains near-optimal results under different norms. A minor novelty is the improvement in the $\ell_2$ case where the existing upper bound for computing well-conditioned coresets was improved from $O(d\log\log d)$ to $O(d)$. A particular point of originality is the application of the proposed algorithm to generate a concrete solution to $\ell_p$ subspace embedding problem, in contrast to existing non-constructive solutions.

**Weaknesses:**

This paper suffers from three major weaknesses: (i) unfocused comparisons of the paper's results to existing literature, (ii) technical issues with the mathematical results, and (iii) the lack of important details in several parts of the paper. Below, I outline some specific instances:

(i.1) [Woodruff and Yasuda, 2023] - How does the notion of “distortion” in Definition 1.2 of [Woodruff and Yasuda, 2023] factor into the analysis of this paper. For example, do the coreset generated by Algorithm have high distortion when applied to the well-conditioned $\ell_2$ coreset problem?

(i.1) Runtime and Solution Size - It is difficult for the reader to find how the proposed algorithm compares to existing literature, as comments are generally scattered throughout the paper. A table that compares runtime and solution size would help in improving this issue.

(ii.1) Proof of Thm. 4.2 - Assuming that indeed $OPT_X \leq (1+\epsilon)d + OPT_T$, the only conclusion is that $vol(MVEE(X)) \leq \exp((1+\epsilon)d) \times vol(MVEE(S))$, which does not coincide with the definition at the beginning of Subsection 2.1, as claimed.

(ii.2) Lemma 3.4 - The analysis seems to break down when $r\leq d+1$, but $r$ does not appear to be restricted in the algorithm (indeed, the “Initialization” discussion following Algorithm 1 seems to suggest that $r=d, d+1$ are valid cases.

(ii.3) Section 3.2 - How do we know that $M$ remains invertible throughout the entire runtime of the algorithm. None of the results (including Lemma 3.3) seem to suggest that $\det M \neq 0$.

(iii.1) Algorithm 1 - What is the significance of the parameter $r$? Can I set it to $d$ to be optimal?

(iii.2) Algorithm 1 - A description of the initialization subroutine and its properties is missing. This is important for the reader verify that the presented algorithm is valid, e.g., $M$ starts with full rank.

(iii.3) Proof of Thm 4.2 - Why is $\sum n_i = r$?

(iii.4) Proof of Thm 4.2 - Why is $H^{-1} = M / r$? Specifically, why is $n_i = 1$ for all $i$ (which is needed for this identity to hold in general).

**Questions:**

Aside from the issues and questions pointed out in the “Weaknesses” section, I have a few minor suggestions/questions below:

(1) Please use either $u_i$ or $\lambda_i$ for the dual variables in the proof of Theorem, but do not use both.

(2) Please make the use of parameter $r$ somewhere in Algorithm 1. At first glance, it appears to be an unused parameter, e.g., setting $r=d$ and $r=1000d$ has no effect on the result of the algorithm.

(3) In equation (4), $r$ can be fractional. This doesn't seem to make sense in the context of the “Initialization” discussion after Algorithm 1.

**Limitations:**

Yes, the authors have sufficiently addressed the limitations of their work.

---

> ### Author Rebuttal · Authors · 2023-08-09
>
> >(i.1) [Woodruff and Yasuda, 2023] - How does the notion of “distortion” in Definition 1.2 of [Woodruff and Yasuda, 2023] factor into the analysis of this paper. For example, do the coreset generated by Algorithm have high distortion when applied to the well-conditioned $\ell_2$ coreset problem?
>
> The notion of distortion in [WY23] is for low rank approximation, it is not related to volumetric spanner / well-conditioned basis, which is the subject of our paper. So in short, the question of distortion does not arise.
>
> >(i.1) Runtime and Solution Size - It is difficult for the reader to find how the proposed algorithm compares to existing literature, as comments are generally scattered throughout the paper. A table that compares runtime and solution size would help in improving this issue.
>
> Thanks for the suggestion, we will add a clear discussion in the final version. Please see our response to Reviewer 5 (4Fat) as well.
>
> >(ii.1) Proof of Thm. 4.2 - Assuming that indeed $OPT_X \le (1+\epsilon)d + OPT_T$, the only conclusion is that $vol(\mathrm{MVEE}(X)) \le \exp((1+\epsilon)d) \times vol(\mathrm{MVEE}(S))$, which does not coincide with the definition at the beginning of Subsection 2.1, as claimed.
>
> This is exactly the definition at the beginning of 2.1 – see the equation there. It is not a “strong coreset”, as we also discuss in Section 2.1.
>
> >(ii.2) Lemma 3.4 - The analysis seems to break down when $r \le d+1$, but $r$ does not appear to be restricted in the algorithm (indeed, the “Initialization” discussion following Algorithm 1 seems to suggest that $r = d, d+1$ are valid cases.)
>
> Yes, $r$ can be any value larger than or equal to d. Note that when $r = d, d+1$, the approximation guarantee is $\Omega(d)$. So, as mentioned in Theorem 3.6, the more interesting values of r are larger, e.g. $r = 3d$ which results in $1$-approximation. Perhaps the reviewer is referring to the denominator being $r-d+1$. This becomes zero when $r = d-1$, not $d+1$.
>
> (If we misunderstood, please refer us to the point in the analysis you think it breaks.)
>
> >(ii.3) Section 3.2 - How do we know that $M$ remains invertible throughout the entire runtime of the algorithm. None of the results (including Lemma 3.3) seem to suggest that $\det M \neq 0$.
>
> Note that the determinant of $M$ does not decrease in the local search update (see condition in line 5). It remains to show that the selected M in the initialization step is invertible. This easily follows from (a) the greedy algorithm is maximizing volume (which is proportional to determinant) and (b) the full set of vectors span a d dimensional space (which was an assumption we can make without loss of generality, as we can always work with the span of the vectors).
>
> For the sake of clarity, we will add this discussion to the “Preliminaries and Notation” section of the paper.
>
> >(iii.1) Algorithm 1 - What is the significance of the parameter $r$? Can I set it to d to be optimal?
>
> As also mentioned in (ii.2), one can set $r$ to any value $\ge d$. However, there is a trade-off between the approximation factor and $r$. When $r = d$ (or $d + o(1)$), the approximation guarantee is $\Omega(d)$.
> Also as mentioned in Theorem 3.6, setting $r = 3d$, the approximation factor becomes one.
>
> >(iii.2) Algorithm 1 - A description of the initialization subroutine and its properties is missing. This is important for the reader to verify that the presented algorithm is valid, e.g., $M$  starts with full rank.
>
> In Algorithm 1, it is mentioned that the pre-processing is described in the text. $M$ is a full rank matrix by the fact that we are maximizing the volume.
>
> >(iii.3) Proof of Thm 4.2 - Why is $\sum n_i = r$?
>
> $n_i$ denotes the number of times index $i$ appears in $S$ (and size of $S$ as a multiset is $r$). So by definition, $\sum n_i = r$.
>
> >(iii.4) Proof of Thm 4.2 - Why is $H^{-1} = M / r$? Specifically, why is $n_i = 1$ for all $i$ (which is needed for this identity to hold in general).
>
> By the definition of $H$, $H^{-1} = \sum_i \lambda_i v_i v_i^T = \sum_i (n_i / r) v_i v_i^T = (1/r) \sum_i n_i v_i v_i^T = M$. In all of this, nowhere is it required that $n_i = 1$. What *is* needed is the invertibility of $H$. This is discussed in the answer to (ii.3) above.
>
> >(1). Please use either $u_i$ or $\lambda_i$ for the dual variables in the proof of Theorem, but do not use both.
>
> Thanks, we will fix this inconsistency.
>
> >(2). Please make the use of parameter r somewhere in Algorithm 1. At first glance, it appears to be an unused parameter, e.g., setting $r = d$ and $r = 1000d$ has no effect on the result of the algorithm.
>
> No, $r$ is used in the Initialization step of the algorithm. The set $S$ is always of size $r$ – so it is indeed very crucial. We will add a note to make this more explicit.
>
> And of course, in Theorem 3.6, both approximation guarantee and runtime are functions of $r$ and $d$. So, $r$ plays a very important role in the final guarantee of the constructed volumetric spanner.
>
> >(3). In equation (4), $r$ can be fractional. This doesn't seem to make sense in the context of the “Initialization” discussion after Algorithm 1.
>
> We will fix the typo and set it to be the ceiling of $(1 + 4/\epsilon) d$

---

> > ### Comment · Reviewer_JkQ2 · 2023-08-10
> > **On (ii.1), (ii.2), (2)**
> >
> > **On (ii.1)**
> >
> > > This is exactly the definition at the beginning of 2.1 – see the equation there. It is not a “strong coreset”, as we also discuss in Section 2.1.
> >
> > No, it is not (note the $\exp(\cdot)$ term in my version).
> >
> > For a more concrete example, consider the case of $\epsilon = 0$ and $d=1$. The definition near the beginning of Subsection 2.1 yields $vol(MVEE(X))\leq vol(MVEE(S))$, but Thm. 4.2 only implies  $vol(MVEE(X))\leq e \cdot vol(MVEE(S)) \approx 2.72 \cdot vol(MVEE(S))$.
> >
> > That is, your algorithm is provably worse than any $e$-approximation algorithm.
> >
> > **On (ii.2)**
> >
> > Yes, you are correct. I meant $r \leq d-1$. Please add a precondition for $r$ in Lemma 3.4 to make this clear.
> >
> > **On (2)**
> >
> > Make sure to do this inside the algorithm as well, to make it more self-contained.

---

> > > ### Author Response · Authors · 2023-08-10
> > > **clarification of typo**
> > >
> > > Ah, sorry about the confusion. This is caused by a typo: the term $(1+\epsilon)d$ should actually be $d \ln (1+\epsilon)$. We argue that $H/(1+\epsilon)$ is a feasible solution (line 311 in the submission). Plugging it into the objective, we have $OPT_X \le -\ln det(H/(1+\epsilon)) = d \ln(1+\epsilon) - \ln det (H)$.  (This is because $det(H/c) = det(H)/c^d$ for a $d$-dimensional $H$, and any constant $c$.)
> > >
> > > We will correct this in the final version, and thank the reviewer for carefully checking.

---

> > > > ### Comment · Reviewer_JkQ2 · 2023-08-12
> > > >
> > > > Thank you for the additional clarifications and responses. I have decided to raise my score from 3 (Reject) to 6 (Weak Accept) in view of them.
> > > >
> > > > Good luck on the rest of your reviews! :)

---

### Official Review · Reviewer_QwvL · 2023-08-01

**Soundness:** 4 excellent
**Presentation:** 4 excellent
**Contribution:** 3 good
**Rating:** 6
**Confidence:** 5

**Summary:**

This paper considers the problem of constructing a volumetric spanner: a subset of vectors which allows to represent every other vector via a linear combination with coefficients whose lp-norm is small. The algorithms are based on local search and work for any lp-norm for p>=1. As a representative result (Theorem 3.6) one can find for a set X of size n>=d a subset of size at most 3d which can be used to represent all vectors in X via linear combinations with l2-norm of the coefficients <= 1. This requires O(d log d) iterations of local search. For l1-norm an exponential lower bound is given (Theorem 3.8). The result for p>2 follows trivially from Theorem 3.6 and for 1<p<2 a construction with certain parameters is given.


**Strengths:**

The paper has clear results and is well-written. I think I understand all the main results and techniques quite well.


**Weaknesses:**

The applications to machine learning problems are somewhat weak (a few examples are given in the intro to column subset selection and sparse coding, but I am not exactly sure what the implications of the results in this paper are for the applications).

The claim that MVEE is an application is that we get an improvement on the size of the best known coreset construction. What is the magnitude of this improvement? Does this yield an algorithmic speedup?

There is an application mentioned to [Woodruff, Yasuda’23] where a log log d factor can be shaved. Can you elaborate on the implications of this as well? The discussion in Appendix A.1 is hard to follow. Can you restate the WY’23 result and explain what the contribution here is?


**Questions:**

– Line 17: sparse coding or problem?
– See a question regarding applications below.


**Limitations:**

Yes

---

> ### Author Rebuttal · Authors · 2023-08-09
>
> > The applications to machine learning problems are somewhat weak (a few examples are given in the intro to column subset selection and sparse coding, but I am not exactly sure what the implications of the results in this paper are for the applications).
>
> Please see our above message to all reviewers for discussion on applications of our results to machine learning.
>
> >The claim that MVEE is an application is that we get an improvement on the size of the best known coreset construction. What is the magnitude of this improvement? Does this yield an algorithmic speedup?
>
> As mentioned in above (message to all reviewers), finding the “optimal” coreset size for MVEE has been studied in several previous works (including Kumar and Yildirim 2007, and Todd 2016). We improve the size from $O(d \log \log d)$ to $O(d)$, using an algorithm of similar complexity. Numerically, this is not much, but it shows that (a) finding min enclosing ellipsoids admits coresets of linear size, similar to much simpler problems, like finding an axis parallel bounding box, and (b) local search offers an improvement over greedy coordinate ascent.
>
> >There is an application mentioned to [Woodruff, Yasuda’23] where a log log d factor can be shaved. Can you elaborate on the implications of this as well? The discussion in Appendix A.1 is hard to follow. Can you restate the WY’23 result and explain what the contribution here is?
>
> [WY23] gives many “black box” applications of well conditioned spanning sets, including entrywise Huber error low rank approximation, oblivious subspace embeddings, etc. In all these works, we can instead use our construction; this gives an improvement by a factor $\log \log d$, and in many cases, gives the tight bounds.
>
> >Q. – Line 17: sparse coding or problem? – See a question regarding applications below.
>
> Thanks, we will fix the typo, it should be “sparse coding problem”. This is only an example where choosing the basis is crucial, it is not directly relevant for us.

---

### Author Rebuttal · Authors · 2023-08-09


**Message to all Reviewers.**

We thank all reviewers for their careful and constructive comments. Here we are addressing the question regarding the application of our result to machine learning and the improvement of our work over prior works. Then, we will answer all other questions in the individual responses. Please let us know if you have any further questions/comments.

**Applications to Machine Learning.**  (The following are mentioned briefly in the introduction, but we can expand on them in the final version.) Well conditioned bases were introduced and studied by Awerbuch and Kleinberg (2008) and Hazan et al. (2013), as a good “exploration basis” for bandit algorithms on convex domains. Bandit optimization is a fundamental and well-studied problem in ML –which itself has many applications– and our results give improvements over these prior work (explained below).

The second main application is matrix low-rank approximation. The problem of finding a small (in cardinality) subset of columns whose combinations can be used to approximate all the other columns of a matrix is called “column subset selection”. It has been used for matrix sketching, “interpretable” low rank approximation, and has applications to streaming algorithms as well.  The classic works of Frieze and Kannan 1997; Drineas, Mahoney and Muthukrishnan 2006; Boutsidis et al. (cited in the paper) all study this question. Well conditioned bases are closely related to column subset selection, and indeed, the works of Boutsidis et al. exploit this connection. The recent paper of Woodruff and Yasuda (STOC 2023) expands on this, using well conditioned bases for a host of matrix approximation problems.

Third, well conditioned bases are closely related to determinantal point processes (DPP) and diversity maximization, also well studied in the ML literature. Indeed, our techniques are derived from work in this space.

Finally, as we point out in the paper, we obtain coresets for the classic problem of min volume enclosing ellipsoids (MVEE). Coresets are an object of extensive study in ML, used to obtain a “representative” sample of a given dataset. Many works have tried to construct “optimal” sized coresets for various data analysis problems, like clustering and low rank approximation. We do this for MVEE, obtaining coresets of size linear in the dimension. This is conceptually interesting, because it matches the coreset size for a much simpler object – the axis-parallel bounding box (which always has a coreset of size $\le 2d$, obtained by taking the two extreme points along each axis).

**Improvement over prior work, significance of the results.** We will highlight the following discussion in the final version.

First off, we note that a unified treatment of the $\ell_p$ well-conditioned bases (for general $p$), along with matching lower bounds has not been done in any of the prior work. Existing results treated the cases $p=2$ and $p=\infty$, using different techniques.

For $p=2$, there are two main prior works. The first is the work of Hazan et al. 2013. They obtain a linear sized basis, similar to our result. However, their result is weaker in terms of running time (by a factor roughly $d^2$), as well as the constants in the size of the basis. Moreover, our algorithm is much simpler, it is simply local search with an appropriate objective, while theirs involves running a spectral sparsification subroutine (that involves keeping track of barrier potentials, etc.) followed by a rounding step.

The second work is the recent result of Woodruff and Yasuda (2023). Here the algorithm is simple – basically a greedy addition (and thus the running time is comparable to ours, up to a logarithmic factor). However, the authors incur an additional $\log \log d$ factor, due to the analysis of the greedy algorithm (which itself was from a prior work of Todd 2016). Removing this factor is interesting conceptually, as it shows that local search with an appropriate objective can achieve something stronger than the best known result for greedy.

---

### Decision · Program_Chairs · 2023-09-21

**Decision:**

Accept (poster)

**Comment:**

Although several reviewers had major technical concerns about the paper, and whether its contributions represented a genuine improvement over previous work, the discussion period resolved these concerns, and nearly all reviewers eventually agreed that the paper should be accepted. The remaining concern is about the novelty of the algorithm, but the authors make a compelling argument that the lack of novelty is itself a discovery, since it demonstrates that an existing and very simple method is nearly optimal, which was not previously known. The authors should consider emphasizing this further in their revision.